# SPOTTING LLMS WITH BINOCULARS: ZERO-SHOT DETECTION OF MACHINE-GENERATED TEXT

## ABSTRACT

Detecting text generated by modern large language models is thought to be hard, as both LLMs and humans can exhibit a wide range of complex behaviors. However, we find that a score based on contrasting two closely related language models is highly accurate at separating human-generated and machine-generated text. Based on this mechanism, we propose a novel LLM detector that only requires simple calculations using pre-trained LLMs. The method, called *Binoculars*, achieves state-of-the-art accuracy without any training data. It is capable of spotting machine text from a range of modern LLMs without any model-specific modifications. We comprehensively evaluate *Binoculars* on a number of text sources and in varied situations. On news documents *Binoculars* detects 94.92% of synthetic samples at a false positive rate of 0.01%, given 512 tokens of text from either humans or ChatGPT, matching highly competitive commercial detectors tuned specifically to detect ChatGPT.

## 1 INTRODUCTION

We present a method for detecting LLM-generated text that works in the zero-shot setting in which no training examples are used from the LLM source. Even with this strict limitation, our scheme still out-performs all open-source methods for ChatGPT detection and is competitive with or better than commercial APIs, despite these competitors using training samples from ChatGPT (Mitchell et al., 2023; Verma et al., 2023). At the same time, because of the zero-shot nature of our detector, the very same detector can spot multiple different LLMs with high accuracy – something that all existing solutions fail to do.

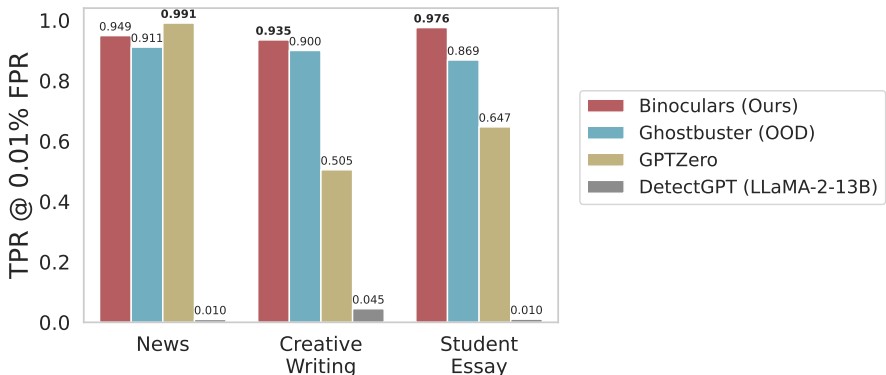

Figure 1: **Detection of Machine-Generated Text from ChatGPT**. Our detection approach using *Binoculars* is highly accurate at separating machine-generated and human-written samples from *News*, *Creative Writing* and *Student Essay* with a false positive rate of 0.01%. *Binoculars*, based on open-source Falcon models with no finetuning, outperforms both commercial detection systems, such as GPTZero, as well as strong open-source detectors – even though both of these baselines are specifically tuned to detect ChatGPT (Verma et al., 2023; Tian, 2023a). Our approach operates entirely in a zero-shot setting and has neither been tuned nor trained to detect ChatGPT in particular.

The ability to detect LLMs in the zero-shot setting addresses issues of growing importance. Prior research on combating academic plagiarism (TurnitIn.com) has fixated strongly on ChatGPT because of its simple and accessible interface. But more sophisticated actors use LLM APIs to operate bots, create fake product reviews, and spread misinformation on social media platforms at a large scale. These actors have a wide range of LLMs available to them beyond just ChatGPT, making zero-shot, model-agnostic detection critical for social media moderation and platform integrity assurance (Crothers et al., 2022; Bail et al., 2023). This zero-shot capability is a departure from existing detectors that rely on model-specific training data and often fail to transfer to new models.

Our proposed detector, called *Binoculars*, works by viewing text through two lenses. First, we compute the $\log$ perplexity of the text in question using an "observer" LLM. Then, we compute all the next-token predictions that a "performer" LLM would make at each position in the string, and compute their perplexity according to the observer. If the string is written by a machine, we should expect these two perplexities to be similar. If it is written by a human they should be different.

## 2 THE LLM DETECTION LANDSCAPE

A common first step in harm reduction for generative AI is detection. Specifically, from documenting and tracing of text origins (Biderman & Raff, 2022) to investigating spam and fake news campaigns (Zellers et al., 2019) and to analyzing training data corpora, classifying text as human or machine generated is common practice (Bender et al., 2021; Crothers et al., 2022; Mirsky et al., 2023).

Successful efforts to spot machine-generated text showed early promise on models whose generation output is not convincingly human. However, with the rise of transformer models for language modeling (Radford et al., 2019; Brown et al., 2020; Chowdhery et al., 2022; Touvron et al., 2023), primitive mechanisms to detect machine-generated text are rendered useless. While one approach is to record (Krishna et al., 2023) or watermark all generated text (Kirchenbauer et al., 2023), these *preemptive detection* approaches can only be implemented with full control over a generation model.

Instead, the recent spread of machine-generated text, especially via ChatGPT, has lead to a flurry of work on *post-hoc detection*, or approaches that can be used to detect machine text without cooperation from model owners. These detectors can be separated into two main groups. The first is trained detection models, where a pretrained language model backbone is finetuned for the binary classification task of detection (Solaiman et al., 2019; Zellers et al., 2019; Yu et al., 2023; Zhan et al., 2023). These techniques include adversarial training (Hu et al., 2023) or abstention (Tian et al., 2023). Instead of finetuning the whole backbone, a linear classifier can also fit on top of frozen learned features, which allows for the inclusion of commercial API outputs (Verma et al., 2023).

The second category of approaches comprises statistical signatures that are characteristic of machine-generated text. These approaches have the advantage of requiring none or little training data and they can easily be adapted to newer model families (Pu et al., 2022). Examples include detectors based on perplexity (Tian, 2023b; Vasilatos et al., 2023; Wang et al., 2023), perplexity curvature (Mitchell et al., 2023), log rank (Su et al., 2023), intrinsic dimensionality of generated text (Tulchinskii et al., 2023), and n-gram analysis (Yang et al., 2023). Our coverage of the landscape is non-exhaustive, and we refer to recent surveys Tang et al. (2023); Dhaini et al. (2023); Guo et al. (2023) as well as our appendix for additional details.

In Section 3, we motivate our approach and discuss why detecting language model text, especially in the ChatGPT world, is difficult. In this work, our emphasis is directed toward baselines that function within post hoc, out-of-domain (zero-order), and black-box detection scenarios. We use state-of-the-art Ghostbuster (Verma et al., 2023), the commercially deployed GPTZero[1], and DetectGPT (Mitchell et al., 2023) to compare detection performance over various datasets in Section 4. In Section 5, we evaluate the reliability of *Binoculars* in various settings that constitute edge cases and interesting behaviors of our detector.

With an understanding of how much work exists on LLM detection, a crucial question arises: How do we appropriately and thoroughly evaluate detectors? Many works focus on accuracy on balanced test sets and/or AUC of their proposed classifiers, but these metrics are not well-suited for

---

[1]https://gptzero.me/

the high-stakes question of detection. Ultimately, only detectors with low false-positive rates across wide distributions of human-written text, truly reduce harm. Further, Liang et al. (2023) note that detectors are often only evaluated on relatively easy in-domain datasets. Their performance on out-of-domain samples is abysmal, for example TOEFL essays written by non-native English speakers were wrongly marked as machine-generated 48-76% of the time by commercial detectors (Liang et al., 2023).

From a theoretical perspective, Varshney et al. (2020), Helm et al. (2023), and Sadasivan et al. (2023) all discuss the limits of detection. These works generally agree that fully general-purpose models of language would be, by definition, impossible to detect. However, Chakraborty et al. (2023) note that even models that are arbitrarily close to this optimum are technically detectable given a sufficient number of samples. In practice, the relative success of detection approaches, such as the one we propose and analyze in this work, provides constructive evidence that current language models are imperfect representations of human writing – and thereby detectable. Finally, the robustness of detectors to attacks attempting to circumvent detection can provide stronger practical limits on reliability in the worst case (Bhat & Parthasarathy, 2020; Wolff & Wolff, 2022; Liyanage & Buscaldi, 2023).

## 3 *Binoculars*: How it works

Our approach, *Binoculars*, is so named as we look at inputs through the lenses of two different language models. It is well known that perplexity – a common baseline for machine/human classification – is insufficient on its own, leading prior work to disfavor approaches based on statistical signatures. However we propose using a ratio of two scores, where one is a perplexity measurement and the other is *cross-perplexity*, a notion of how surprising the next token predictions of one model are to another model. This two-model mechanism is the basis for our general and accurate detector, and we show that this mechanism is able to detect a number of large language models, even when they are unrelated to the two models used in the *Binoculars*.

### 3.1 Background & Notation

A string of characters $s$ can be parsed into tokens and represented as a list of token indices $\vec{x}$. Let $x_i$ denote the token ID of the $i$-th token, which refers to an entry in the LLMs vocabulary $V = \{1, 2..., n\}$. Given a token sequence as input, a language model $\mathcal{M}$ predicts the next token by outputting a probability distribution over the vocabulary:

$$\mathcal{M}(T(s)) = \mathcal{M}(\vec{x}) = Y$$
$$Y_{ij} = \mathbb{P}(v_j | x_{0:i-1}) \text{ for all } j \in V. \tag{1}$$

We will abuse notation and abbreviate $\mathcal{M}(T(s))$ as $\mathcal{M}(s)$ where the tokenizer is implicitly the one used in training $\mathcal{M}$. For our purposes, we define $\log \text{PPL}$, the log-perplexity, as the average log-likelihood of all tokens in the given sequence. Formally, let

$$\log \text{PPL}_{\mathcal{M}}(s) = -\frac{1}{L} \sum_{i=1}^{L} \log(Y_{ix_i}), \tag{2}$$

where $\vec{x} = T(s)$, $Y = \mathcal{M}(\vec{x})$ and $L =$ number of tokens in $s$

Intuitively, log-perplexity measures how "surprising" a string is to a language model. As mentioned above, perplexity has been used to detect LLMs, as humans produce more surprising text than LLMs. This is reasonable, as $\log \text{PPL}$ is also the loss function used to train generative LLMs, and models are likely to score their own outputs as unsurprising. For narrative purposes, it is convenient to discuss the *perplexity of a string according to some model*.

Our method also measures how surprising the output of one model is to another. We define the *cross-perplexity*, which takes two models and a string as its arguments. Let $\log \text{X-PPL}_{\mathcal{M}_1, \mathcal{M}_2}(s)$ measure the average per-token cross-entropy between the outputs of two models, $\mathcal{M}_1$ and $\mathcal{M}_2$ ,

when operating on the tokenization of $s$.[2]

$$\log \text{X-PPL}_{\mathcal{M}_1, \mathcal{M}_2}(s) = -\frac{1}{L} \sum_{i=1}^{L} \mathcal{M}_1(s)_i \cdot \log\left(\mathcal{M}_2(s)_i\right) \tag{3}$$

Note that $\cdot$ denotes the dot product between two vector-valued quantities. Intuitively, the cross perplexity measures how surprising one language model's predictions are to another observer model.

## 3.2 What makes detection hard? A primer on the capybara problem.

Why do we require measurements of both perplexity and cross-perplexity? Unsurprisingly, LLMs tend to generate text that is unsurprising. Meanwhile, because humans differ from machines, human text has high perplexity to an LLM observer. For this reason, it is tempting to use raw perplexity for LLM detection.

Unfortunately, this intuition breaks when hand-crafted prompts are involved. Prompts have a strong influence over downstream text, and prompts are typically unknown to the detector. On the one hand, the prompt "1, 2, 3," might result in the very low perplexity completion "4, 5, 6." On the other hand, the prompt "Can you write a few sentences about a capybara that is an astrophysicist?" will yield a response that seems more strange. In the presence of the prompt, the response may be unsurprising. In the absence of the prompt, a response containing the curious words "capybara" and "astrophysicist" in the same sentence will have high perplexity, resulting in the false determination that the text was written by a human, see the example in Table 1. Clearly, certain contexts will result in high perplexity and others low perplexity, regardless of whether the author is human or machine. We refer to this dilemma as "the capybara problem" – in the absence of the prompt, LLM detection seems difficult.

> "Dr. Capy Cosmos, a capybara unlike any other, astounded the scientific community with his groundbreaking research in astrophysics. With his keen sense of observation and unparalleled ability to interpret cosmic data, he uncovered new insights into the mysteries of black holes and the origins of the universe. As he peered through telescopes with his large, round eyes, fellow researchers often remarked that it seemed as if the stars themselves whispered their secrets directly to him. Dr. Cosmos not only became a beacon of inspiration to aspiring scientists but also proved that intellect and innovation can be found in the most unexpected of creatures." - ChatGPT

Table 1: This quote is LLM output from ChatGPT (GPT-4) when prompted with "Can you write a few sentences about a capybara that is an astrophysicist?" The Falcon LLM assigns this sample a high perplexity (2.20), well above the mean for both human and machine data. Despite this problem, our detector correctly assigns a *Binoculars* score of 0.73, which is well below the machine/human threshold of 0.84, resulting in a correct classification with high confidence. For reference, DetectGPT wrongly assigns a score of 0.14, which is below the optimal threshold of 0.17, and classifies the text as human. GPTZero assigns a 49.71% score that this text is generated by AI.

## 3.3 Our Detection Score

*Binoculars* solves the capybara problem by providing a mechanism for estimating the baseline perplexity induced by the prompt. By comparing the perplexity of the observed text to this expected baseline, we get fiercely improved LLM detection.

**Motivation** Language models are known for producing low-perplexity text relative to humans and thus a perplexity threshold classifier makes for an obvious detecting scheme. However, in the LLM era, the generated text may exhibit a high perplexity score in the absence of the prompt (see the "Capybara Problem" in Table 1). To calibrate for prompts that yield high-perplexity generation, we use *cross-perplexity* introduced Equation (3) as a normalizing factor that roughly encodes the perplexity level of next-token predictions from two models.

---

[2]This requires that $\mathcal{M}_1$ and $\mathcal{M}_2$ share a tokenizer.

Rather than examining raw perplexity scores, we instead propose measuring whether the tokens that appear in a string are surprising *relative to the baseline perplexity of an LLM acting on the same string.* A string might have properties that result in high perplexity when completed by any agent, machine or human. Yet, we expect the next-token choices of humans to be even higher perplexity than those of a machine. By normalizing the observed perplexity by the expected perplexity of a machine acting on the same text, we can arrive at a detection metric that is fairly invariant to the prompt; see Table 1.

We propose the *Binoculars* score $B$ as a sort of normalization or reorientation of perplexity. In particular we look at the ratio of perplexity to cross-perplexity.

$$B_{\mathcal{M}_1, \mathcal{M}_2}(s) = \frac{\log \text{PPL}_{\mathcal{M}_1}(s)}{\log \text{X-PPL}_{\mathcal{M}_1, \mathcal{M}_2}(s)} \qquad (4)$$

Here, the numerator is simply the perplexity, which measures how surprising a string is to $\mathcal{M}_1$. The denominator measures how surprising the token predictions of $\mathcal{M}_2$ are when observed by $\mathcal{M}_1$. Intuitively, we expect a human to diverge from $\mathcal{M}_1$ more than $\mathcal{M}_2$ diverges from $\mathcal{M}_1$, provided the LLMs $\mathcal{M}_1$ and $\mathcal{M}_2$ are more similar to each other than they are to a human.

The *Binoculars* score is a general mechanism that captures a statistical signature of machine text. We will see that, for most obvious choices of $\mathcal{M}_1$ and $\mathcal{M}_2$, it does separate machine and human text much better than perplexity alone. Importantly, it is capable of detecting generic machine-text generated by neither model $\mathcal{M}_1$ nor $\mathcal{M}_2$.

Interestingly, we can draw some connection to other approaches that contrast two strong language models, such as contrastive decoding (Li et al., 2023), which aims to generate high-quality text completions by generating text that roughly maximizes the difference between a weak and a strong model. Speculative decoding is similar (Chen et al., 2023; Leviathan et al., 2023), it uses a weaker model to plan completions. Both approaches function best when pairing a strong model with a very weak secondary model. However, as we show below, our approach works best for two models that are very close to each other in performance. In the remainder of this work, we use the open-source models Falcon-7b model ($\mathcal{M}_1$) and the Falcon-7b-instruct ($\mathcal{M}_2$) (Almazrouei et al., 2023).

## 4 ACCURATE ZERO-SHOT DETECTION

In this section we evaluate our proposed score, and build a zero-shot LLM detector with it. With *Binoculars*, we are able to spot machine-generated text in a number of domains. In our experimental evaluation, we focus on the problem setting of detection of machine-generated text from a modern LLM, as generated in common use cases without consideration for the detection mechanism.

### 4.1 DATASETS

We start our experiments with several datasets described in the LLM detection literature. The most recent baseline to which we compare is Ghostbuster. Verma et al. (2023), who propose this method, introduce three datasets that we include in our study: *Writing Prompts*, *News*, and *Student Essay*. These are balanced datasets with equal numbers of human samples and machine samples. The machine samples are written by ChatGPT.

We also generate several datasets of our own to evaluate our capability in detecting other language models aside from ChatGPT. Drawing samples of human-written text from CCNews (Hamborg et al., 2017), PubMed (Sen et al., 2008), and CNN (Hermann et al., 2015), we generate alternative, machine-generated completions using LLaMA-2-7B and Falcon-7B. To do so, we peel off the first 50 tokens of each human sample and use it as a prompt to generate up to 512 tokens of machine output. We then remove the human prompt from the generation and only use the purely machine-generated text in our machine-text datasets. Further, we use the Orca dataset (Lian et al., 2023), which provides several million instruction prompts with their machine-generated completions from chat versions of GPT-3 and GPT-4. This dataset allows us to check the reliability of the proposed method when detecting instruction-tuned models, and allows us to quantify detection differences between GPT-3 and GPT-4.

## 4.2 METRICS

Since detectors are binary classifiers, the standard suite of binary classification metrics are relevant. In particular, it is often considered comprehensive to look at ROC curves and use the area under the curve (AUC) as a performance metric. In fact, Verma et al. (2023) and Mitchell et al. (2023) only report performance as measured by AUC and F1 scores. We argue that these metrics alone are inadequate when measuring LLM detection accuracy.

In high-stakes detection settings, the most concerning harms often arise from *false positives*, i.e., instances when human text is wrongly labeled as machine-generated. For this reason, we focus on true-positive rates (TPR) at low false-positive rates (FPR), and adopt a standard FPR threshold of $0.01\%$.[3] We will present F1 scores and AUC values only for comparison to prior publications, but we prefer to focus on TPR values at low FPR as a key metric. The reader may observe that AUC scores are often uncorrelated with TRP@FPR when the FPR is below $1\%$.

## 4.3 BENCHMARK PERFORMANCE

Using a handful of datasets, we compare the AUC and TPR of *Binoculars* to Ghostbuster (Verma et al., 2023), GPTZero (Tian, 2023a), and DetectGPT (using LLaMA-2-13B to score curvature) (Mitchell et al., 2023). We highlight that this is a comparisons on machine samples from ChatGPT are *in favor* of GPTZero and Ghostbuster, as these detectors have been tuned to detect ChatGPT output, and comparisons using samples from LLaMA models are *in favor* of DetectGPT for the same reason.

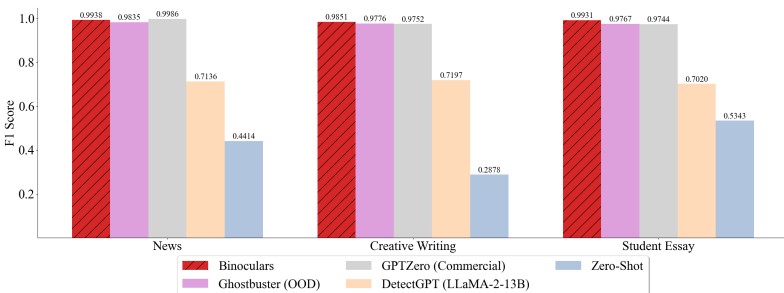

Figure 2: F1 scores for detection of ChatGPT-generated text indicate that several detectors perform similarly. We discuss below how this metric can be a poor indicator of performance at low FPR.

Our score-based detector needs only a threshold to separate machine and human text, which we preset using reference data. To maintain our "zero-shot" claim, we set the threshold using the combination of training splits from all of our reference datasets: News, Creative Writing, and Student Essay datasets from Verma et al. (2023), which are generated using ChatGPT. We also compare detectors on LLaMA-2-13B and Falcon-7B generated text with prompts from CC News, CNN, and PubMed datasets. All of these datasets have an equal number of human and machine-generated text samples. We optimize and fix our threshold globally using these datasets. As one exception, to be sure that we meet the Ghostbuster definition of "out-of-domain," when comparing our performance with Ghostbuster we do not include the ChatGPT datasets (News, Creative Writing, and Student Essay) in the threshold determination, and only use samples from CC News, CNN, and PubMed (generated via LLaMA and Falcon) to choose our threshold.

**Ghostbuster Datasets.** The Ghostbuster detector is a recent detector tuned to detect output from ChatGPT. Using the same three datasets introduced and examined in the original work by Verma et al. (2023), we show in Figure 2 that *Binoculars* outperforms Ghostbuster in the "out-of-domain" setting. This setting is the most realistic, and includes evaluation on datasets other than Ghostbuster's training data. A desirable property for detectors is that with more information they get stronger. Figure 3 shows that both *Binoculars* and Ghostbuster have this property, and that the advantages of *Binoculars* are even clearer in the few-token regime.

---

[3]The smallest threshold we can comprehensively evaluate with our limited compute resources.

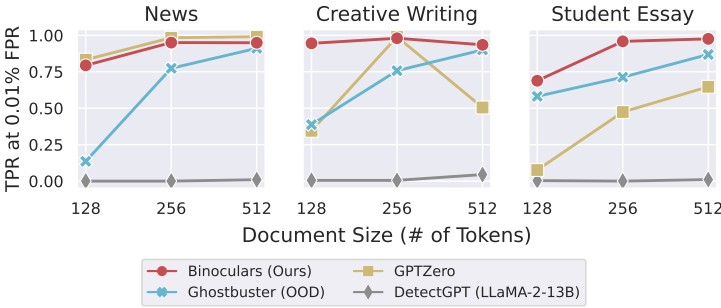

Figure 3: **Document Size Impact on Detection Performance**. The plot displays the TPR at 0.01% FPR across varying document sizes. The x-axis represents the number of tokens of the observed document, while the y-axis indicates the corresponding detection performance, highlighting the *Binoculars* ability to detect with a low number of tokens.

**Open-Source Language Models.** We show that our detector is capable of detecting the output of several LLMs, such as LLaMA as shown in Figure 4 and Falcon as shown in Figure 13 in the appendix. Here we also observe that Ghostbuster is indeed only capable of detecting ChatGPT generation and fails to reliably detect LLaMA generated text. We use AUC plot to compare threshold invariant performance for all methods.

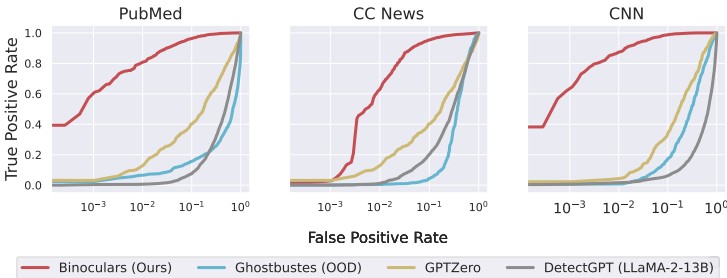

Figure 4: **Detecting LLaMA-2-13B generations.** *Binoculars* achieves higher TPR for low FPR than competing baselines.

**Orca Data**. The Orca dataset contains machine generations from both GPT-3 and GPT-4 for a wide range of tasks (Lian et al., 2023). This serves as a diverse test bed for measuring *Binoculars* on both of these modern and high-performing LLMs. Impressively, *Binoculars* detects 92% of GPT-3 samples and 89.57% of GPT-4 samples when using the F1-optimal threshold (from reference datasets). Note, we only report accuracy since this is over a set of machine-generated text only.

## 5 RELIABILITY IN THE WILD

How well does *Binoculars* work when faced with scenarios encountered in the wild? We comprehensively evaluate on memorized samples, text from non-native speakers, modified prompting strategies, and other edge cases in this section.

### 5.1 VARIED TEXT SOURCES

To explore detector performance in even more settings, we also investigate the Multi-generator, Multi-domain, and Multi-lingual (M4) detection datasets (Wang et al., 2023). These samples come from Arxiv, Reddit, Wikihow, and Wikipedia sources, and include examples in varied languages, such as Urdu, Russian, Bulgarian, and Arabic. Machine text samples in this dataset are generated via ChatGPT. In Figure 7, we show the precision and recall of *Binoculars* and four other baselines, showing that our method generalizes across domains and languages. These baselines, released

with M4 Datasets, include Logistic Regression over Giant Language Model Test Room (LR GLTR) (Gehrmann et al., 2019) which generates features assuming predictions are sampled from predicted token distribution, Stylistic (Li et al., 2014) which employs syntactic features at character, word and sentence level, News Landscape classifiers (NELA) (Horne et al., 2019) which generates and leverages semantic and structural features for veracity classification. Results with more source models appear in Figure 5.

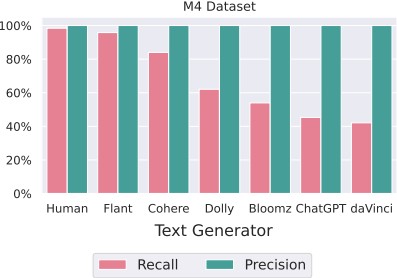

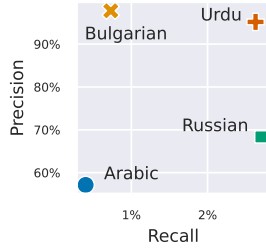

Figure 5: Performance of *Binoculars* on samples from various generative models from the M4 Dataset.

Figure 6: *Binoculars* operate at high precision in Bulgarian and Urdu, while observing low recall in all four languages.

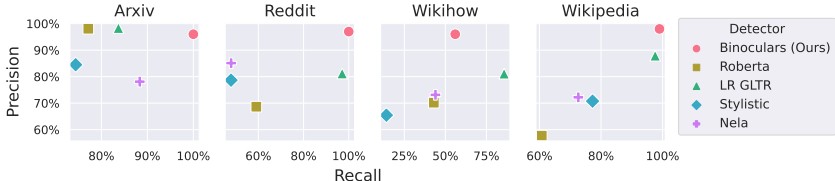

Figure 7: **Detection of ChatGPT-generated text in various domains from M4 Dataset.** Binoculars maintain high precision over 4 domains using the global threshold (tuned out-of-domain) for detection. We use the mean of out-of-domain performance metrics reported by Wang et al. (2023)

## 5.2 OTHER LANGUAGES

When evaluating *Binoculars* on sample from languages that are not largely present in Common Crawl data (standard LLM pretraining data), we find that false-positives rates remain rather low. However, even machine text in these other languages is classified as human. Figure 6 shows that we have reasonable precision but poor recall in these settings. (A stronger multilingual pair of models would likely lead to a version of *Binoculars* that can see ChatGPT-generated text in these languages reliably.) This finding suggests that it may be more accurate to consider *Binoculars* as a detector for LLM output rather than a human-machine classifier. This subtle difference aligns with the performance on samples in languages that most LLMs trained on Common Crawl data cannot generate as well as the findings on randomized data and on memorized examples above.

**English text written by non-native speakers.** A significant concern about LLM detection algorithms, as raised in Liang et al. (2023), is that LLM detectors are inadvertently biased against non-native English speakers (ESL) classifying their writing exceedingly often as machine-generated. To test this, we analyze essays from *EssayForum*, a web page for ESL students to improve their academic writing (EssayForum, 2022). This dataset contains both the original essays, as well as grammar-corrected versions. We compare the distribution of *Binoculars* scores across the original and the grammar-corrected samples. Interestingly, and in stark comparison to commercial detectors examined by Liang et al. (2023) on a similar dataset, *Binoculars* attain equal accuracy at 99.67% (see Figure 8) for both corrected and uncorrected essay datasets. We also point out that *Binoculars* distribution of non-native English speaker's text highly overlaps with that of grammar-corrected versions of the same essays, showing that detection through *Binoculars* is insensitive to this type of shift.

## 5.3 MEMORIZATION

One common feature of perplexity-based detection is that memorized examples are also classified as machine-generated. For example, famous quotes that appear many times in the training data likely have low perplexity even though they are human samples. By looking at several examples, we examine how *Binoculars* performs on this type of data.

First, we ask about the US Constitution – a famous document on which modern LLMs are great at sentence completion. This example has a *Binoculars* score of 0.76, well into the machine range – perhaps less than ideal, but of the 11 famous texts we study,[4] this was the lowest score (most machine-y), and only three fall on the machine-side of our threshold.

It is important to note that while this behavior may be surprising, and does require careful consideration in deployment, it is fully consistent with a machine-text detector. Memorized text is in an interesting category, as it is both text written by human writers, and text that is likely to be generated by an LLM.

Classification of memorized text as machine generated may be acceptable or even desirable in some applications (e.g., plagiarism detection), or undesirable in others (e.g., removal of LLM-generated text from a training corpus). Thus, these experiments highlight the care one needs to take when deploying detectors in practice.

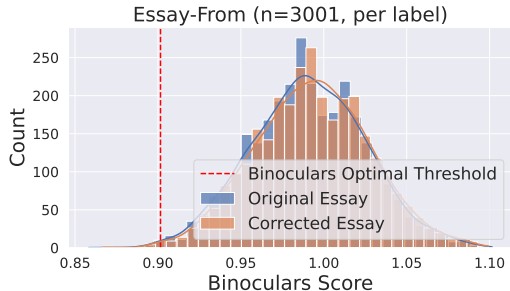

Figure 8: The distribution of *Binoculars* scores remains unchanged when the English grammar is corrected in essays composed by non-native speakers. Both uncorrected and corrected essays are unambiguously classified as human-written.
.

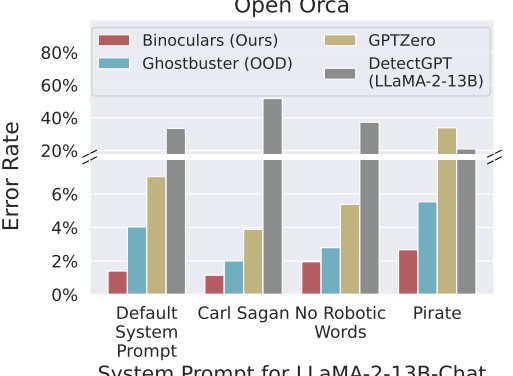

Figure 9: **Modified Prompting Strategies** LLaMA-2-13B-chat with a modified system prompt for text generation.

## 5.4 MODIFIED PROMPTING STRATEGIES

Simple detection schemes are sometimes fooled by simple changes to prompting strategies, which can produce stylized text that deviates from the standard output distribution. With this in mind, we use LLaMA-2-13B-chat and prompts designed to tweak the style of the output. Specifically, we prompt LLaMA2-13B-chat with three different system prompts by appending to the standard system prompt a request to write like a pirate, like Carl Sagan, or without any mechanical or robotic sounding words. The biggest impact we observe arises when asking for pirate-sounding output and this only increases the error rate by 1%; see Figure 9.

## 5.5 RANDOMIZED DATA

We want to test whether arbitrary mistakes, hashcodes, or other kinds of randomized errors will bias the model towards false positives. To test the impact of randomness, we generate random sequences of tokens from the Falcon tokenizer, and score them with our *Binoculars* as usual. We plot histograms for this distribution in Fig. 16. We find that this distribution is assigned an abnormally high score, with a mean around 1.35 for Falcon, far beyond the range of human text (which has a mean of around 1).

---

[4]See table 4 in appendix A.2 for all famous text evaluated under Binoculars.

This behavior is the opposite of what we observed above; while memorized text is classified as machine-generated, with generally lower scores, randomized text is classified with a score exceeding human text. This is expected, as trained LLMs are strong models of language and exceedingly unlikely to replicate a completely random sequence of tokens in any situation.

## 6    DISCUSSION AND LIMITATIONS

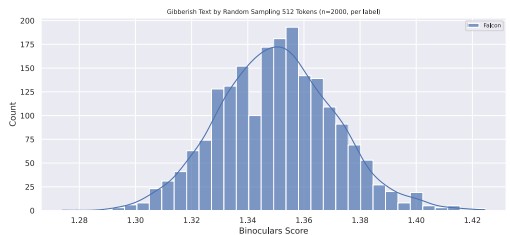

Figure 10: Score distribution when sampling random tokens from the Falcon tokenizer.

We present *Binoculars*, a method for detecting LLMs without training data. Our transferable detector works in the zero-shot setting, without access to the particular model used for generation or example data from it. We speculate that this transferability arises from the similarity between modern LLMs, as they all use nearly identical transformer components and are likely trained on datasets comprising mostly Common Crawl (commoncrawl.org) data from similar time periods. As the number of open source LLMs rapidly increases, the ability to detect multiple LLMs with a single detector is a major advantage of *Binoculars* when used in practice, for example for platform moderation.

Our study has a number of limitations. Due to limited GPU memory, we could not perform extensive studies with larger (30B+) open-source models. Further, we focus on the problem setting of detecting machine-generated text in normal use, and we do not consider explicit efforts to bypass detection. Finally, we also do not have access to sufficient data at this time to perform further evaluation on non-conversational domains, such as source code.

Overall, we propose a simple detection strategy based on a two-model mechanism using cross-perplexity that is surprisingly capable of detecting generic machine-generated context, regardless of which exact LLM generated the text, and without tuning.

## REPRODUCIBILITY STATEMENT

We provide details on all datasets used, and on the exact method that we employ in the main body of this work. Additionally, we provide code to exactly replicate our detection score implementation with the supplementary material of this work. We note that comparisons to commercial detection APIs, such as GPTZero are based on API evaluations from September 2023, and may not be reproducible in the future, underscoring the importance of transparent, open-source solutions.

## ETHICS STATEMENT

Language model detection may be a key technology to reduce harm, whether to monitor machine-generated text on internet platforms and social media, filter training data, or identify responses in chat applications. Nevertheless, care has to be taken so that detection mechanisms actually reduce harm, instead of proliferating or increasing it. We provide an extensive reliability investigation of the proposed *Binoculars* mechanisms in Section 5, and believe that this is a significant step forward in terms of reliability, for example when considering domains such as text written by non-native speakers. Yet, we note that this analysis is only a first step in the process of deploying LLM detection strategies and does not absolve developers of such applications from carefully verifying the impact on their systems. We especially caution that the existence of LLM detectors does not imply that using them is worthwhile in all scenarios.

Also, we explicitly highlight that we consider the task of detecting "naturally" occurring machine-generated text, as generated by LLMs in common use cases. We understand that no detector is perfect and we do not guarantee any performance in settings where a motivated adversary tries to fool our system. We present a thorough evaluation across a wide variety of test sources, but we maintain that directed attempts to bypass detection might be possible as is often the case.

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

# A Appendix

## A.1 Ablation Studies

**Comparison to Other Model Pairs.**

Table 2: Other combinations of scoring models, evaluated on our reference datasets as described in the main body.

| PPL Scorer ($\mathcal{M}_1$) | X-Cross PPL Scorers ($\mathcal{M}_1'$, $\mathcal{M}_2$) | TPR at 0.01% FPR | TPR at 0.1% FPR | F1-Score | AUC |
|---|---|---|---|---|---|
| Falcon-7B-Instruct | Falcon-7B, Falcon-7B-Instruct | 100.0 | 100.0 | 1.0 | 1.0 |
| Llama-2-13B | Llama-13B, Llama-2-13B | 99.6539 | 99.6539 | 0.9982 | 0.9999 |
| Llama-2-7B | Llama-7B, Llama-2-7B | 99.3079 | 99.3079 | 0.9965 | 0.9998 |
| Llama-2-13B | Llama-13B, Llama-2-13B | 98.3549 | 98.3549 | 0.9913 | 0.9997 |
| Falcon-7B-Instruct | Falcon-7B, Falcon-7B-Instruct | 98.72 | 99.16 | 0.9953 | 0.9996 |
| Falcon-7B-Instruct | Falcon-7B, Falcon-7B-Instruct | 94.92 | 99.4 | 0.9963 | 0.9996 |
| Llama-2-7B | Llama-7B, Llama-2-7B | 95.8441 | 97.5757 | 0.9922 | 0.9996 |
| Llama-2-13B | Llama-13B, Llama-2-13B | 98.64 | 99.04 | 0.9953 | 0.9995 |
| Llama-2-7B | Llama-7B, Llama-2-7B | 98.8 | 99.28 | 0.9959 | 0.9995 |
| Llama-2-7B | Llama-7B, Llama-2-7B | 98.16 | 98.6 | 0.9937 | 0.9992 |
| Llama-2-13B | Llama-13B, Llama-2-13B | 98.4 | 98.72 | 0.9943 | 0.9992 |
| Falcon-7B-Instruct | Falcon-7B, Falcon-7B-Instruct | 94.1125 | 97.922 | 0.9926 | 0.9992 |
| Falcon-7B-Instruct | Falcon-7B, Falcon-7B-Instruct | 93.5 | 93.5 | 0.9875 | 0.9990 |
| Falcon-7B-Instruct | Falcon-7B, Falcon-7B-Instruct | 92.0 | 92.0 | 0.9918 | 0.9990 |
| Llama-2-7B | Llama-7B, Llama-2-7B | 94.0 | 94.0 | 0.9850 | 0.9989 |
| Llama-2-7B | Llama-7B, Llama-2-7B | 98.0 | 98.0 | 0.9956 | 0.9988 |
| Falcon-7B-Instruct | Falcon-7B, Falcon-7B-Instruct | 72.6957 | 72.7857 | 0.9908 | 0.9988 |
| Llama-2-13B | Llama-13B, Llama-2-13B | 97.875 | 97.875 | 0.9931 | 0.9987 |
| Llama-2-13B-Chat | Llama-2-13B, Llama-2-13B-Chat | 71.3199 | 82.6799 | 0.9846 | 0.9986 |
| Llama-2-13B | Llama-13B, Llama-2-13B | 97.5 | 97.5 | 0.9875 | 0.9985 |
| Falcon-7B-Instruct | Falcon-7B, Falcon-7B-Instruct | 97.5778 | 97.5778 | 0.9930 | 0.9983 |
| Falcon-7B-Instruct | Falcon-7B, Falcon-7B-Instruct | 23.3076 | 48.3732 | 0.9842 | 0.9975 |
| Llama-2-13B | Llama-13B, Llama-2-13B | 0.32 | 32.08 | 0.9840 | 0.9968 |
| Llama-2-13B-Chat | Llama-2-13B, Llama-2-13B-Chat | 20.9172 | 60.0671 | 0.9763 | 0.9968 |
| Llama-2-13B | Llama-13B, Llama-2-13B | 47.1476 | 69.2953 | 0.9747 | 0.9964 |

**String Length.** Is there a correlation between *Binoculars* score and sequence length? Such correlations may create a bias towards incorrect results for certain lengths. In Figure 11, we show the joint distribution of token sequence length and *Binocular* score. Sequence length offers little information about class membership.

**Score Components.** Perplexity is used by many detecting formulations in isolation. We show in Figure 12 that both perplexity and cross-perplexity are not effective detectors in isolation.

## A.2 Other famous texts

Two songs by Bob Dylan further demonstrate this behavior. *Blowin' In The Wind*, a famous Dylan track has a much lower Falcon perplexity than his unreleased song *To Fall In Love With You* (PPL values are 1.11 and 3.30, respectively.) It might be reasonable for famous songs to get classified as machine text and they are more likely output than less famous songs. *Binoculars*, however, labels both of these samples confidently as human samples (with scores of 0.92, and 1.01, respectively).

## A.3 Distribution over various datasets

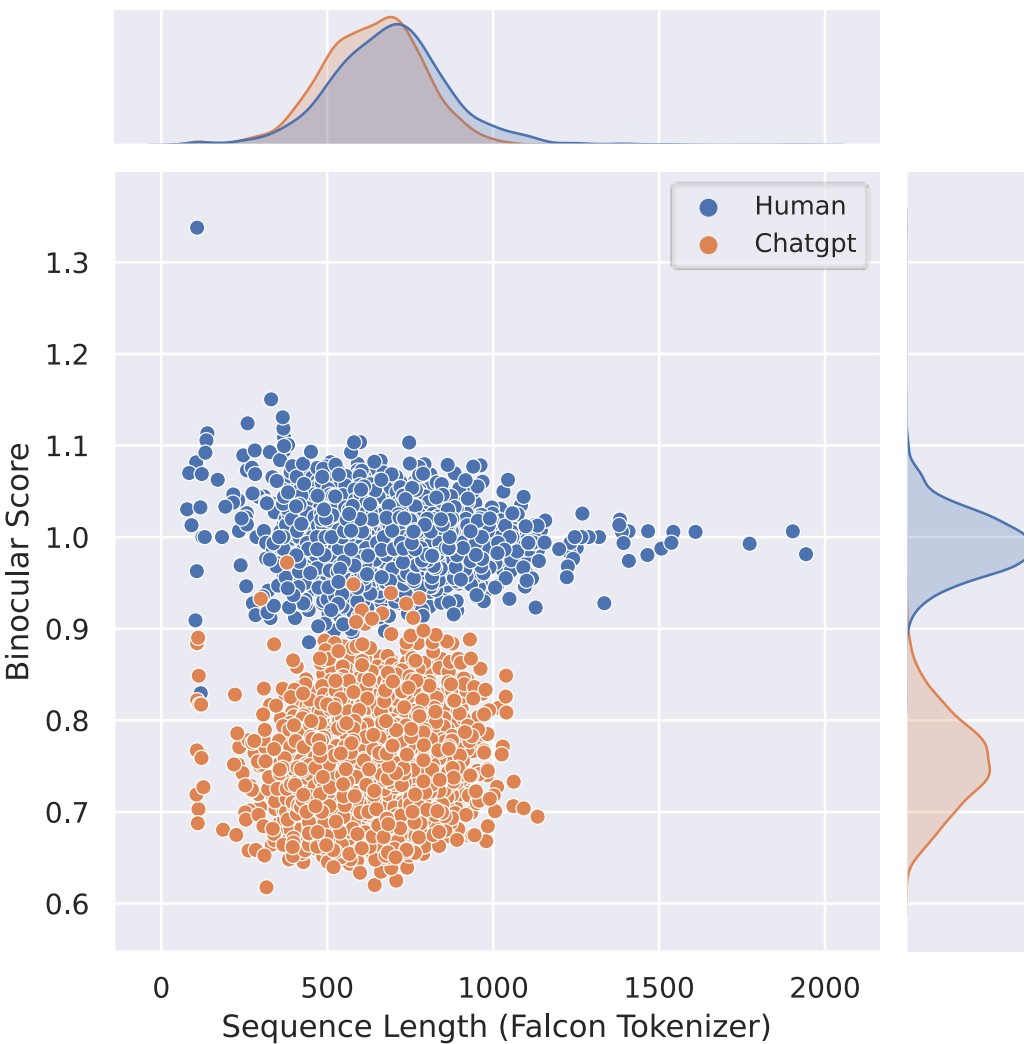

Figure 11: A closer look at the actual distribution of scores in terms of sequence length for the Ghostbuster news dataset.

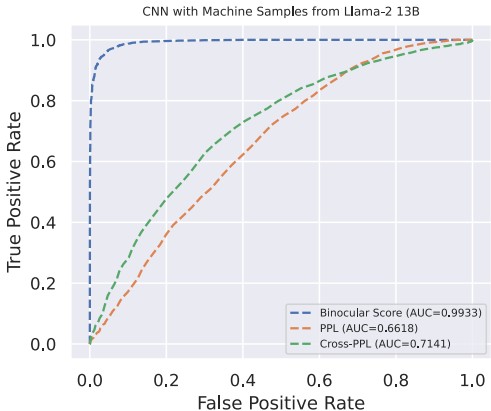

Figure 12: Perplexity and Cross-perplexity are not strong detectors on their own.

Table 3: Over various datasets, we show that perplexity alone or cross-perplexity alone are poor predictors of human versus machine, whereas Binoculars perform well even at low false-positive rates (FPR).

| Dataset | Detector | AUC | True Positive Rate | | | |
|---|---|---|---|---|---|---|
| | | | @ 0.01% FPR | @ 0.1% FPR | @ 1% FPR | @ 5% FPR |
| Writing Prompts | Falcon PPL | 1.00 | 0.86 | 0.86 | 0.94 | 0.98 |
| | Falcon X-PPL | 0.94 | 0.56 | 0.56 | 0.59 | 0.79 |
| | LLaMA PPL | 0.99 | 0.86 | 0.86 | 0.92 | 0.98 |
| | LLaMA X-PPL | 0.86 | 0.04 | 0.04 | 0.10 | 0.43 |
| | Binoculars-Falcon | **1.00** | 0.93 | 0.93 | 0.96 | **1.00** |
| | Binoculars-LLaMA | **1.00** | **0.95** | **0.95** | **0.98** | **1.00** |
| News | Falcon PPL | 0.99 | 0.65 | 0.77 | 0.90 | 0.95 |
| | Falcon X-PPL | 0.85 | 0.04 | 0.12 | 0.29 | 0.53 |
| | LLaMA PPL | 0.98 | 0.67 | 0.71 | 0.89 | 0.95 |
| | LLaMA X-PPL | 0.26 | 0.00 | 0.00 | 0.00 | 0.01 |
| | Binoculars-Falcon | **1.00** | 0.95 | **0.99** | **1.00** | **1.00** |
| | Binoculars-LLaMA | **1.00** | **0.99** | **0.99** | **1.00** | **1.00** |
| Essay | Falcon PPL | 1.00 | 0.78 | 0.78 | 0.88 | 0.99 |
| | Falcon X-PPL | 0.93 | 0.25 | 0.25 | 0.38 | 0.70 |
| | LLaMA PPL | 0.99 | 0.42 | 0.42 | 0.90 | 0.98 |
| | LLaMA X-PPL | 0.80 | 0.01 | 0.01 | 0.04 | 0.16 |
| | Binoculars-Falcon | **1.00** | 0.98 | 0.98 | 0.99 | **1.00** |
| | Binoculars-LLaMA | **1.00** | **0.99** | **0.99** | **1.00** | **1.00** |

Table 4: Case Studies of Text Samples likely to be memorized by LLMs.

| Human Sample | PPL (Falcon 7B Instruct) | Cross PPL (Falcon 7B, Falcon 7B Instruct) | Binoculars Score | Predicted as Human-Written |
|---|---|---|---|---|
| US Constitution | 0.6680 | 0.8789 | 0.7600 | ✗ |
| "I have a dream speech" | 1.0000 | 1.2344 | 0.8101 | ✗ |
| Snippet from Cosmos series | 2.3906 | 2.8281 | 0.8453 | ✗ |
| Blowin' In the Wind (song) | 1.1172 | 1.2188 | 0.9167 | ✓ |
| Oscar Wilde's quote | 2.9219 | 3.0781 | 0.9492 | ✓ |
| Snippet from White Night | 2.6875 | 2.8125 | 0.9556 | ✓ |
| Wish You Were Here | 2.5000 | 2.5938 | 0.9639 | ✓ |
| Snippet from Harry Potter book | 2.5938 | 2.6875 | 0.9651 | ✓ |
| First chapter of A Tale of Two Cities | 2.7188 | 2.7500 | 0.9886 | ✓ |
| Snippet from Crime and Punishment | 2.8750 | 2.9063 | 0.9892 | ✓ |
| To Fall In Love With You (song) | 3.2969 | 3.2656 | 1.0096 | ✓ |

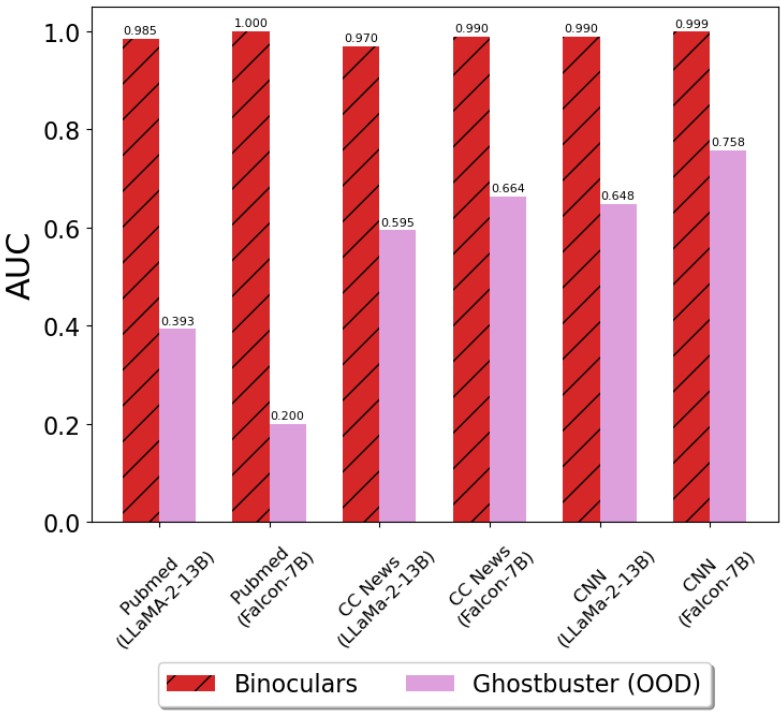

Figure 13: Comparison of Ghostbuster and Binoculars AUC on PubMed, CCNews and CNN datasets.

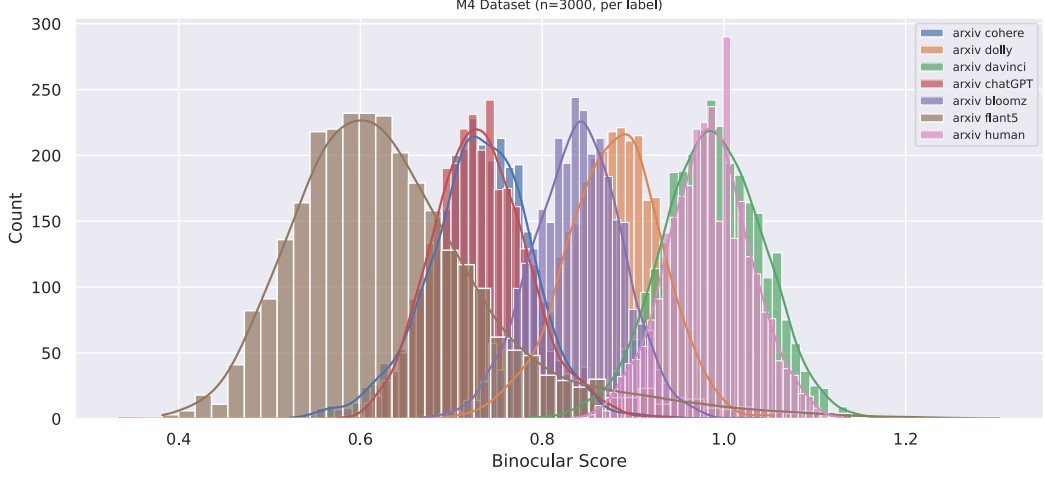

Figure 14: Binoculars Score on generations based on arxiv documents

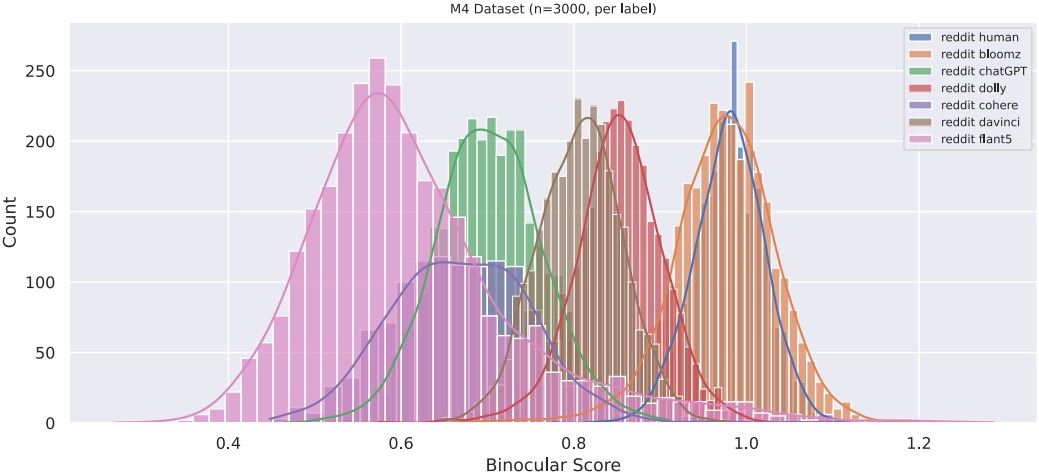

Figure 15: Binoculars Score on generations based on Reddit documents

## A.4 RANDOMIZED DATA

We want to test whether arbitrary mistakes, hashcodes, or other kinds of randomized errors will bias the model towards false positives. To test the impact of randomness, we generate random sequences of tokens from the Falcon tokenizer, and score them with our *Binoculars* as usual. We plot histograms for this distribution in Fig. 16. We find that this distribution is assigned an abnormally high score, with a mean around 1.35 for Falcon, far beyond the range of human text (which has a mean of around 1).

This behavior is the opposite of what we observed above; while memorized text is classified as machine-generated, with generally lower scores, the randomized text is classified with a score exceeding human text. This is expected, as trained LLMs are strong models of language and exceedingly unlikely to replicate a completely random sequence of tokens in any situation.

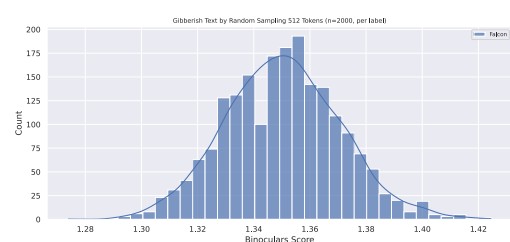

Figure 16: Score distribution when sampling random tokens from the Falcon tokenizer.

## A.5 IDENTICAL SCORING MODEL

We inspect Binocular's performance when we choose to use identical $\mathcal{M}_1$ and $\mathcal{M}_1$ models in equation (4). We use Falcon-7B and Falcon-7B-Instruct models and compare the two performances with Binoculars Score over dataset by (Verma et al., 2023) in Fig. 17. We observe although the vanilla Binoculars score is best over 3 domains, using Falcon-7B as input models is competitive.

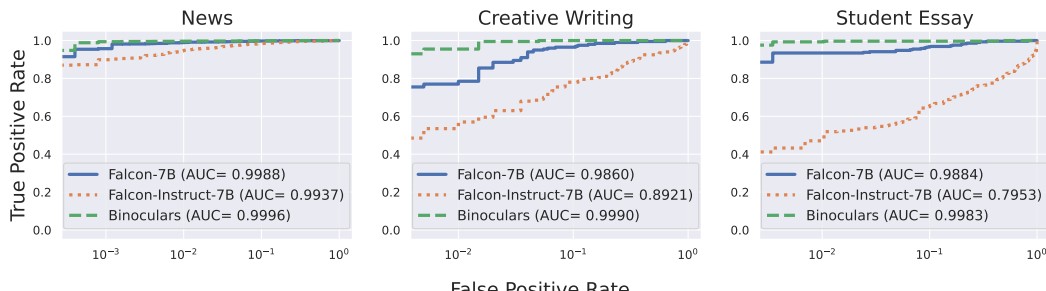

Figure 17: **AUC Curve** Binoculars score using identical $\mathcal{M}_1$ and $\mathcal{M}_2$ models using Falcon-7B and Falcon-7B-Instruct.

