# OpenReview forum: "Spotting LLMs With Binoculars: Zero-Shot Detection of Machine-Generated Text"
_ICLR.cc/2024/Conference — Submitted to ICLR 2024_

### Official Review · Reviewer_CMmZ · 2023-10-31

**Soundness:** 1 poor
**Presentation:** 2 fair
**Contribution:** 2 fair
**Rating:** 3
**Confidence:** 3

**Summary:**

The paper describes a method to detect text generate by an LLM. The method can be applied to text generated by any LLM without fine-tuning and it is based on the perplexity of the text according to a an observer LLM, but normalized by the perplexity of the text generated by another LLM using the same input string, The method is evaluated on several public datasets and compared to some other machine-generated detectors.

**Strengths:**

The proposed method is simple and the rationale behind the proposed normalized perplexity is well motivated and seems to be appropiate. It does not require any specific training and can be used with text generated by any LLM and also with any LLM to compute the perplexity.

**Weaknesses:**

I have several concerns regarging the experimental validation and the presentatation of the results:
- In section 4.2 it is argued that TPR@FPR is a better metric than F1 score or AOU, and particularly TPR at 0.01% of FPR. However TPR at 0.01 FPR is never used in the results presented afterwards and mainly F1, AUC, precision and recall are used in most of the figures. Only in figures 4 and 6 TPR vs FPR is plot, but without specifically analyzing TPR at 0.01% FPR as claimed in section 4.2. Related to this, in the abstract it is claimed that "On news documents Binoculars detect 95% of synthetic samples at a false positive rate of 0.01%". In the results only figure 4 for the student essay dataset seems to show a result similar to this one, but it is very specific to only of the datasets used in the experiments.
- Evaluation should be done in a more systematic and coherent way among different datasets to be able to better compare the performance of the proposed method with SoA. As long as it is possible the same methods should be used for comparison. However, the set of methods used to compare with in figures 2, 4, 5 and 6 are different, and even methods used in figure 6 are not mentioned or discussed in the text.
- The selection of the methods used to compare with is not clearly motivated. In section 2 many methods are described and categorized in different categories. I would be nice to have a systematic comparison with more methods of each of the categories defined in section 2.
- I miss an ablation study analyzing some of the choices made in the design of the method. For instance, the contribution of using 	the proposed detection score vs. simple perplexity or the effect of the LLMs used both to compute perplexity and to generate the text to be classified. Is there any difference in performance (positive or negative) if the observer LLM is the same LLM used to generate the text under analysis?
- It is not clear what figure 3 is showing. Also, in the caption of figure 5 it is said that the comparison is on LLaMa text while in the text referring to the figure it is mentioned text generated by LLaMa, but also by Falcon. Not clear exactly whaat figure 5 is showing.
- There is no comparison of the results obtained in the Orca dataset with results obtained by other methods.
- The description of M4 datasets in section 4.3 would fit better in section 4.1 with the description of the rest of the datasets.

And finally, a minor comment with respect equation (3): shouldn't it be log(M_2(T(s))_i) instead of log(M_2(T(s))_j)?

**Questions:**

See above in weaknesses

---

> ### Author Response · Authors · 2023-11-21
> **Response to Reviewer CMmZ (Part 1)**
>
> We thank the reviewer for their valuable time and effort in providing this constructive review. We used collective feedback from reviewers and conducted additional experiments with results in Figure 3 (per-token detection performance), Figure 4 (open-source generation detection), Figure 9 (detecting different prompting strategies), and Figure 17 (ablation). Below is our response with a point-by-point note for the questions/weaknesses listed.
>
> Firstly, we highlight that the “cross-perplexity” definition is a novel concept in NLP to the best of our knowledge. Its usage in perplexity-detectors for normalization is motivated in section 3.3, resulting in a significant increase in performance by addressing the perplexity-only-detectors limitations (as shown in Figure 17 in Appendix A5). The definition and use of x-PPL came about after viewing the perplexity-only-detector (and its limitation)  from a specific lens as mentioned in “Capybara problem” (section 3.2) important for motivating this work.
>
> Based on the feedback, we have updated the draft highlighting our changes with blue text.
>
> __1. Usage of TPR @ low-FPR across experiments__
> - In the metrics section, we discuss how detectors’ performance should be evaluated under a low-false-positive regime (which can be realized by many metrics) since type 1 errors are significantly more costly than type 2 in downstream tasks. We also argue that typically reported metrics like standalone AUC values, as in other baselines, don’t translate into real-life performance sufficiency (as shown in Table 2: AUCs are uncorrelated with TPR at low FPR). Below is our rationale for metrics in figures:
>   - Figure 1 reports TPR @ 0.01% FPR and Figure 2 reports F1-Score on the identical dataset as Figure 1 (ChatGPT benchmark by [1])
>   - Figure 3 is updated to have TPR @ 0.01% FPR at the varying sizes of samples (# of tokens)
>   - Figure 4 reports TPR observed at different FPR levels (on log scale) for LLaMA-2-13B generations over 3 datasets
>   - Figure 5, 6, and 7 reports how Binoculars maintain high precision (with varying recall) over multiple generators, languages and domains respectively.
>   - Figure 9's dataset contains only machine-generated samples and hence only reports an error rate.
>
> __2. The abstract claim of detecting news documents Binoculars detect 95% of synthetic samples at a false positive rate of 0.01%__
> - This figure is from the first column in Figure 1 (News, ChatGPT dataset [1]). We updated the abstract to replace the previously rounded number to 94.92%.
>
> __3. Document size impact figure only contains the Student dataset__
> - We updated Figure 3 to include all 3 datasets from [1] and also added another baseline method (GPTZero).
>
> __4. Performance comparison with state-of-the-art and using the same methods should be used for comparison__
> - Our emphasis is directed toward baselines that function within post hoc, out-of-domain (zero-order), and black-box detection scenarios. We use state-of-the-art Ghostbuster (Verma et al., 2023), the commercially deployed GPTZero, and DetectGPT to compare detection performance over various datasets in Section 4.
> - Performance Benchmark: As per above, on all benchmark datasets in section 4, we use these baselines (which include the ChatGPT benchmark dataset competing baseline in [1]). To align our experiments better, we have updated the figure as mentioned above in this note. We thank you for your feedback on this.
> - Reliability: In Section 5, we evaluate the reliability of Binoculars in various settings that constitute edge cases and interesting behaviors of our detector. With the exception of Figure 9 (modified prompting strategies), we report these numbers as absolute to aide understanding of Binoculars specifically.
>
> __5. Methods in Figure 6 are not mentioned or discussed in the text__
> - Thank you for this feedback. We have added a paragraph in section 5.1 for these methods.
>
> __6. The selection of the methods used to compare with is not motivated.__
> - We updated the draft to motivate our choice of baselines (also mentioned in the previous point) (section 2). To reiterate, we focus on evaluating baselines designed for post hoc, out-of-domain (zero-order), and black-box detection scenarios. which includes state-of-the-art Ghostbusters (Verma et al., 2023), the commercially deployed GPTZero, and DetectGPT.
>
> __7. Ablation study to ascertain the contribution of using the proposed detection score vs. simple perplexity__
> - Due to the page limit we have this study in Figure 12 + Table 3 in Appendix A. We compare PPL, xPPL, and Binoculars (Falcon and LLaMA variants) performance over the ChatGPT dataset from 1]. We see how Binoculars perform better than its components to detect machine text.

---

> > ### Author Response · Authors · 2023-11-21
> > **Response to Reviewer CMmZ (Part 2)**
> >
> > __8. The difference in performance (positive or negative) if the observer LLM is the same LLM used to generate the text under analysis__
> > - Figure 13 includes performance when using Falcon-7B (which powers the Binoculars score) for sample generation. We do not see a remarkable change in performance or a visible trend when using Falcon-7 B-produced text.
> >
> > __9. The caption of Open Source (LLaMA generated) detection says the comparison is on LLaMa text while the text referring to the figure mentions LLaMA and Falcon__
> > - The results for Falcon generations are available in Figure 13 in Appendix A3. We have fixed the error in text and now both main Figure 4 and appendix Figure are referenced correctly.
> >
> > __10. Not clear what Figure 3 (performance by document length) shows__
> > - This plot shows the performance of our detector at different levels of document length (measured by # of tokens at 128, 256 and 512). We also added 2 more datasets and an additional baseline in GPTZero to strengthen our claim of high performance while seeing smaller passage of text.
> >
> > __11. The description of M4 datasets in section 4.3 would fit better in section 4.1 with the description of the rest of the datasets.__
> > -  In Section 4 we use baseline methods to compare our performance with established zero-shot, OOD domain detectors on various known benchmark datasets. Our intent for M4 datasets is to study our detector's performance in multiple domains, languages, and text generators and report non-relative claims.
> > - Constraint: Unlike other feedback, we could not run other methods on M4 datasets which consist of 39 datasets with 3k samples each due to sizable constraints: a) Ghostbusters require approx $1150 OpenAI credits to get log-probs for 46 million tokens (512 per sample), b) computational constraint for DetectGPT which requires computing multiple perturbations.
> > - Thus, we move M4 datasets experiments to Section 5 (reliability in domain, language, text-generators, etc.).
> >
> > __12. There is no comparison of the results obtained in the Orca dataset with results obtained by other methods__
> > - We run other methods and append baselines in Figure 9 to show our method is more robust than other methods against different system prompting strategies.
> >
> > __13. Typo in formulation/notation.__
> > - We thank you for the detailed feedback and have updated to fix all pointers.
> >
> >
> >
> > To summarize, we present a novel detector that is able to maintain high efficacy in a false-positive regime under various domains/datasets/baselines, using only open-source components to detect in zero-shot settings. Broadly, we highlight the need to evaluate detectors in terms of low-false positive metrics and showcase how our method is performative in this regime (note: in principle, by lowering the threshold even further we can virtually eliminate any false positives). We further study our detector’s reliability in settings that constitute edge cases to understand interesting behaviors, abilities, and limitations of our detector.
> >
> > We are happy to take on any follow-up questions or other feedback.
> >
> > Again, we thank you for your time and contribution to this key area of secure and safe machine learning in the generative AI era.
> >
> > [1] Vivek Verma, Eve Fleisig, Nicholas Tomlin, and Dan Klein. Ghostbuster: Detecting Text Ghostwritten by Large Language Models. arxiv:2305.15047[cs], May 2023. doi: 10.48550/arXiv.2305. 15047. URL http://arxiv.org/abs/2305.15047.

---

> ### Author Response · Authors · 2023-11-21
> **A Gentle Reminder**
>
> Dear Reviewer CMmZ,
>
> As the discussion period ends, we would like to kindly ask for your reply to our remarks. We feel we have addressed your concerns and we look forward to further discussions and guidance. Please let us know if there is anything else we can do or other questions we can answer.

---

> > ### Comment · Reviewer_CMmZ · 2023-11-22
> >
> > Dear authors,
> > thank you for your detailed response, addressing most of my concerns. I do not require further clarifications. I will carefully review your response along with the other reviewer's comments before making my final recommendation.

---

### Official Review · Reviewer_M521 · 2023-11-01

**Soundness:** 3 good
**Presentation:** 4 excellent
**Contribution:** 3 good
**Rating:** 8
**Confidence:** 3

**Summary:**

This paper presents a simple method to detect machine-generated text, based on measures of perplexity of two independent LMs.
The score is derived from the assumption that a texts generated by two LMs are more similar with each other than with human text.
A comprehensive evaluation is carried out to show that the proposed method has a high detection rate at a low false positive rate.
This remains true for various models used for generating texts. The proposed method gives competitive or superior results compared
to several other methods. Finally, the authors discuss the potential limitations of their method when used in practice

**Strengths:**

The paper is well written, easy to follow and to understand.
The proposed method looks very simple and quite easy to implement. It is well evaluated and seems to give good results, which is a nice combination.
It extends perplexity-based method in a simple and well motivated way.
The paper makes a number of good/interesting points (e.g. the motivation to measure the True Positive Rate at low FPRs, the remark on memorization/randomization in Section 5, the one on LLM similarity).
Most questions that arise when reading the paper are actually answered in the appendix (e.g. how about using different LLM for the score computation, how does the text length affect the detection).
The discussion of limitations in sections 5 and 6 is particularly appreciated: it is still too rarely seen in papers. They naturally raise questions about the potential weaknesses of the approach but the transparency is valuable.

**Weaknesses:**

The paper is quite nice in terms of contents, presentation and evaluation of the proposed method for the 10-page limit.

Further analysis of some aspects could add value to the presented work, although I acknowledge that not everything can fit in the paper (some of which will be asked in the "Questions" section:

  - In section 2, it could be clearer how the proposed method relates to other PPL-based ones
  - In Table 2, we see that the performance can vary a lot depending on the chosen LLM for scoring: what makes a good LLM for the method? Why other choices reach lower TPR?
  - The point about similarity of LLMs is interesting. What would happen if the LM (evaluated or chosen for scoring) is less similar, e.g. different training set, different kind of model, etc.

Minor remarks:

  - in 5.1, mention that Table 4 is in the appendix
  - 4.3 "Fig. 3" -> Figure 3

**Questions:**

Most of the questions I had while reading the manuscript were answered in the appendix.

Some other aspects that I would be curious about following the read of this paper:

  - The point made at the end of 5.1 is interesting but it is not so clear what should be concluded from that remark. It also begs the question of how would the training set of the LLMs (mostly written by humans I guess) be detected
  - The point made in 5.4 (randomization) is interesting as well. It would be interesting to see if the detector could be fooled by randomly changing "some" words in the generated text. I guess the remark made in 5.1 about the desirability of outcome is relevant here too (= how to consider human-edited machine-generated text)
  - The point of 5.2 is interesting and we could also wonder how robust this detection method is. For example, is it easy to tweak existing LLMs to fool the detector (other than prompt), or during training, how easy would it be to integrate the detection score to fool the detector.

---

> ### Author Response · Authors · 2023-11-21
> **Response to Reviewer M521**
>
> We thank the reviewer for their valuable time and effort in providing this constructive review. We are glad that you found our paper well-written, and easy to follow. We used collective feedback from reviewers and conducted additional experiments with results in Figure 3 (per-token detection performance), Figure 4 (open-source generation detection), Figure 9 (detecting different prompting strategies), and Figure 17 (ablation). Below is our response with a point-by-point note for the questions/weaknesses listed.
>
> __1. In section 2, it could be clearer how the proposed method relates to other PPL-based ones__
> - One difference between Binoculars and some other perplexity-based detectors is in the number of forward passes required to compute a discriminative score. For example, DetectGPT is built on a similar motivation -- that perplexity alone is insufficient for classification -- but to overcome the weakness in the signal, DetectGPT utilizes the perplexity of several perturbations to the input. In addition to the overhead of generating perturbations, it also requires computing many perplexities and thus it can be far more computationally expensive. So there may be various ways to extract meaningful information from perplexity, ours is less computationally demanding than some others.
>
> __2. In Table 2, we see that the performance can vary a lot depending on the chosen LLM for scoring: what makes a good LLM for the method? Why other choices reach lower TPR?__
> - This is more because of how stringent TPR @ 0.01% FPR as a performance metric is than instability of performance. We updated the table 2 to include TPR @ 0.1% FPR. F1-Score and AUC. As you may see, the performances are "stable" through the lens of F1-Score and AUC - which are fine metrics to use when the costs of type 1 and 2 errors are uniform.
> - However, in problems with the severely skewed cost of error towards FP (like detection), metrics like (vanilla) F1-score and AUC don’t provide downstream safety in the deployment of these systems.
> - We also note (in section 4.2) “that AUC scores are often uncorrelated with TRP@FPR when the FPR is below 1%.” (as seen in the updated Table 2).
> - Moreover, we see saturation of performance using Binoculars (at least in its current format) irrespective of the scorers used. However, with the surge in open-source LMs, we do not claim an exhaustive search of scoring models to be used.
>
> __3. What would happen if the LM (evaluated or chosen for scoring) is less similar, e.g. different training set, different kind of model, etc.__
> - Please note, by our x-PPL definition, we have an implicit constraint of having to use two models to share a tokenizer, and most available open-source models have a high degree of training set overlap (eg. Falcon, Llama families, etc.).
> - The cross-perplexity measures the degree to which two models' next-token distributions overlap. We suspect cross-ppl from two models trained on dissimilar datasets would provide more noise than the signal we want to detect (see updated text section 3.3 for motivation of x-PPL).
> - The similar behavior of the transformers model is what we depend on for cross-ppl and Binoculars score definition. Thank you for mentioning this important point. We expanded the definition section to mention the impact of different training sets and will be happy to provide any more information required.
> - In the ablation experiment (in Figure 17), we see how x-PPL and PPl alone aren't enough for reliable detection performance.
>
> __4. How would the training set of the LLMs (mostly written by humans I guess) be detected__
> - Training samples are a super-set of memorized strings in Section 5.3 (i.e. strings that were part of training but may or may not be re-generated by LLM).
> - For a perplexity-based detector, memorized/famous text is more difficult to classify in comparison to samples from larger training sets (since the famous text is expected to be in the training set multiple times).
> - Also, the classification of such samples is domain-specific since both humans and machines can be deemed to have produced it in different domains (eg. plagiarism detection v/s  removal of LLM-generated text from a training corpus).
> - With this understanding and performance on memorized text, we believe that our detector would score samples from the training set towards the human side of the threshold and should be leveraged accordingly in downstream tasks.

---

> > ### Author Response · Authors · 2023-11-21
> > **Response to Reviewer M521 (Part 2)**
> >
> > __5. Can the detector be fooled by randomly changing "some" words in the generated text?__
> > - Our focus is on detecting machine-generated text in typical usage scenarios. We specifically exclude consideration of explicit efforts to bypass detection, such as adversarial perturbed text detection using Binoculars (or any variant).
> > - However somewhat related, in section 5.3, we observe that Binoculars are insensitive to grammar-corrective changes in human-generated text (same accuracy for either text). Specifically, in Figure 8 we see a comparison of the two distributions of Binoculars.
> > - Adversarial evasion/attack on Binoculars is left for future work.
> >
> > __6. Is it easy to tweak existing LLMs to fool the detector (other than prompt), or during training, how easy would it be to integrate the detection score to fool the detector?__
> > - In this work, we concentrate on detecting unadulterated text from language models and leave the adversarial attack on Binoculars to future work.
> > - As such [1] discusses the theoretical limits of detection and highlights that with enough paraphrasing, detection can be impossible.
> > - For example, LMs can be hard-prompted, soft-prompted, or even fine-tuned to produce text that evades detection or particular detector.
> > - There can be several such attack models used to bypass detection which we leave for future work.
> >
> > To summarize, we present a novel detector that is able to maintain high efficacy in a false-positive regime under various domains/datasets/baselines, using only open-source components to detect in zero-shot settings. Broadly, we highlight the need to evaluate detectors in terms of low-false positive metrics and showcase how our method is performative in this regime (note: in principle, by lowering the threshold even further we can virtually eliminate any false positives). We further study our detector’s reliability in settings that constitute edge cases to understand interesting behaviors, abilities, and limitations of our detector.
> >
> > We are happy to take on any follow-up questions or other feedback.
> >
> > Again, we thank you for your time and contribution to this key area of secure and safe machine learning in the generative AI era.
> >
> > [1] Vinu Sankar Sadasivan, Aounon Kumar, Sriram Balasubramanian, Wenxiao Wang, and Soheil Feizi. Can AI-Generated Text be Reliably Detected? arxiv:2303.11156[cs], March 2023. doi: 10. 48550/arXiv.2303.11156. URL http://arxiv.org/abs/2303.11156.

---

> ### Author Response · Authors · 2023-11-21
> **A Gentle Reminder**
>
> Dear Reviewer M521,
>
> As the discussion period ends, we would like to kindly ask for your reply to our remarks. We feel we have addressed your concerns and we look forward to further discussions and guidance. Please let us know if there is anything else we can do or other questions we can answer.

---

### Official Review · Reviewer_iAnY · 2023-11-02

**Soundness:** 2 fair
**Presentation:** 2 fair
**Contribution:** 2 fair
**Rating:** 3
**Confidence:** 4

**Summary:**

To detect text generated by large language models, the authors propose in this work a method called Binoculars. Different from previous methods for separating human-generated and machine-generated text, Binoculars utilize two models instead of one, to compute two scores: perplexity and cross-perplexity. The ratio of perplexity to cross-perplexity is defined as the Binocular score. The proposed does not need training examples and works in the zero-shot setting.

**Strengths:**

1. The idea of using two large language models to compute a score to distinguish between human-generated and machine-generated text is novel and seems to be effective to certain extent.
2. The paper is overall well-written, and the core idea and technical details are clearly presented.

**Weaknesses:**

1. The experiments are insufficient in that: (1) The experiments and comparisons in Figure 5 are crucial, but the authors only compared with Ghostbuster and the analyses are too brief; (2) According to the ablation study in A.1 in the APPENDIX, the authors actually only conducted experiments using the models from the Falcon and Llama-2 families. Why other types of open-source or closed-source large language models (such as ChatGPT, GPT-4 and Baichuan 2) are not adopted?
2. The reason behind the effectiveness of the proposed Binoculars method is not fully explained.

**Questions:**

The authors should resolve the questions in the Weaknesses section.

---

> ### Author Response · Authors · 2023-11-21
> **Response to Reviewer iAnY**
>
> We thank the reviewer for their valuable time and effort in providing this constructive review. Based on the feedback, we have updated the draft highlighting our changes with blue text. We used collective feedback from reviewers and conducted additional experiments with results in Figure 3 (per-token detection performance), Figure 4 (open-source generation detection), Figure 9 (detecting different prompting strategies), and Figure 17 (ablation). Below is our response with a point-by-point note for the questions/weaknesses listed.
>
> __1. Only Ghostbusters used as a baseline in open-source language models text detection (LLaMA-2-13B)__
> - We add baseline experiments in Figure 4. The performances on Falcon-7B generations can be found in Figure 13 in Appendix A3.
>
> __2. Only models from Falcon and Llama-2 families used for computing x-PPL__
> - Thanks for noting this. This is because of the following:
> - Implicit constraint of x-PPL computation which requires two models to have identical tokenizer (we mention in section 3.1)
> - As per our intuition (and motivation, section 3.3), our detector's performance is dependent on having two _similar_ (by training set) models for computing x-PPL and would only cause a marginal change in performance across different scoring model pairs. We show this in the updated Table 2 in Appendix A1, how F1-Score and AUC metrics (which weigh type 1 and type 2 error equally) are stable across multiple pairs of scoring models only resulting in a marginal change in the performance of the detector. Please note, that we do not claim our scoring models search to be exhaustive.
>
> __3. The reason behind the effectiveness of Binoculars__
> - In the updated draft, we develop on motivation behind our detector (use of x-PPL) in section 3.3: "Language models are known for producing low-perplexity text relative to humans and thus a perplexity threshold classifier makes for an obvious detecting scheme. However, in the LLM era, the generated text may exhibit a high perplexity score in the absence of the prompt (“Capybara Problem” in Table 1). To calibrate for prompts that yield high-perplexity generation, we use cross-perplexity introduced in Equation (3) as a normalizing factor that roughly encodes the perplexity level of next-token predictions from two models."
> - We employ this cross-perplexity, which encodes the degree with which two models' next-token distributions overlap, as our normalizing mechanism to solve the "Capybara Problem." The similar behavior of the transformers model is what we depend on for the x-PPL and Binoculars score definition. Thank you for mentioning this important point. We expanded the definition section to mention the impact of different training sets and will be happy to provide any more information required.
>
> To summarize, we present a novel detector that is able to maintain high efficacy in a false-positive regime under various domains/datasets/baselines, using only open-source components to detect in zero-shot settings. Broadly, we highlight the need to evaluate detectors in terms of low-false positive metrics and showcase how our method is performative in this regime (note: in principle, by lowering the threshold even further we can virtually eliminate any false positives). We further study our detector’s reliability in settings that constitute edge cases to understand interesting behaviors, abilities, and limitations of our detector.
>
> We are happy to take on any follow-up questions or other feedback.
>
> Again, we thank you for your time and contribution to this key area of secure and safe machine learning in the generative AI era.

---

> ### Author Response · Authors · 2023-11-21
> **A Gentle Reminder**
>
> Dear Reviewer iAnY,
>
> As the discussion period ends, we would like to kindly ask for your reply to our remarks. We feel we have addressed your concerns and we look forward to further discussions and guidance. Please let us know if there is anything else we can do or other questions we can answer.

---

### Official Review · Reviewer_FmjG · 2023-11-02

**Soundness:** 2 fair
**Presentation:** 2 fair
**Contribution:** 2 fair
**Rating:** 3
**Confidence:** 3

**Summary:**

This paper presents a novel machine-generated text detector that is based on a simple metric that can be obtained from existing pre-trained LLMs. The proposed metric (Binoculars score) is computed as the ratio between the perplexity and cross-perplexity of a given sample text for two pre-trained LLMs. Using the Binoculars score they build a simple threshold-based classifier to separate machine-generated and human text.

**Strengths:**

+ The proposed Binoculars score for machine-generated text detection can be computed from pre-trained LLM models without any re-training/finetuning.
+ The obtained results are promising on a variety of machine-generated text detection scenarios, including text generated with different generators, and in a variety of domains.

**Weaknesses:**

The authors claim their method is a zero-shot detector, but they "optimize and fix the detection threshold globally using these datasets". The fact that the used pre-trained LLMs are not retrained/fine-tuned does not imply the proposed method is zero-shot, as they optimize the detection threshold for the target task with (in most of the cases) some in-domain data.

I have several concerns regarding the experimental section of this paper.

- Several of the baseline models are not explained in the text: Zero-shot (in Figure 1) and Roberta, LR GLTR, Stylistic, Nela (in figure 6).
- The comparison with baselines is not consistent across experiments, in some Figures/Tables the proposed method is compared with a set of baselines while in others the baselines are different. The choice of baselines seems quite trivial and makes it difficult to assess the contribution of proposed method:
- In Figure 5, Ghostbuster (Verma et al., 2023) is not trained for this dataset, while the detection threshold of the roposed method seems to be optimized on it. Why no other baselines are shown in this plot?
- The results on the M4 Datasets (Figures 3 and 6) are not well explained in my opinion and seem to be not consistent with the results in (Wang et al., 2023). In (Wang et al., 2023) the authors present results on different settings: same-generator cross-domain experiments, same-domain cross-generator experiments. In here it is not specified whether the numbers shown in Figure 6 come from one or the other setting, and the values provided for the baselines' results do not match (at least for what I can appreciate) with the ones found in the tables of (Wang et al., 2023).
- The numbers in Figure 3 are not compared with any baselines. This comparison would be highly relevant due to the authors claim on being able to detect text generated with any text-generator.
- For the Orca Dataset no baselines' results are provided. Same for all the experiments in section 5. Not having any baseline results on these experiments makes it very difficult to quantify the quality of the proposed solution.

When showing FPR/TPR plots, how are these plots generated? by changing the detection threshold? I do not understand why you say 0.01% FPR threshold is "The smallest threshold we can comprehensively evaluate with our limited compute resources."


Apart from all this, it seems to me that the formulation/notation of the Binoculars score is confusing in some aspects.

- $L$ (in eq. 2 and 3) is not defined anywhere in the text, I assume it is the sentence-length.
- The subindex $j$ in eq. 3 is not defined. (Maybe it is a typo?)
- In eq.3 it would be better to use $\overrightarrow{x}$ (as defined before) instead of $T(s)$. It would make the equation more readable.
- If I'm not missing something, the measure expressed in eq. 3 is the $log XPPL$ not $XPPL$. At least it looks consistent with the $log PPL$ definition in eq.2.
- Although the authors first define $log PPL$ in eq. 2, they use $PPL$ instead in the formulation of the Binocular score in eq. 4.

Moreover, since cross-perplexity ($XPPL$) is not a standard term in NLP or machine learning, I would expect to see a bit more of discussion on its interpretation and its effects on the proposed Binoculars score: what are the upper/lower bounds of B? what happens when M_1 and M_2 are the same pre-trained model? what if they are trained in two totally different domains? I appreciate the effort made in section 3.2 (Capybara problem) for the case of "hand-crafted prompts", but other aspects of the Binoculars score should be discussed as well.

**Questions:**

Please clarify those aspects mentioned in the weaknesses section of my review that do not imply new experiments: zero-shot claim, choice of baselines, Binoculars score formulation and interpretation, etc.

---

> ### Author Response · Authors · 2023-11-20
> **Response to Reviewer FmjG (Part 1)**
>
> We thank the reviewer for their valuable time and effort in providing this constructive review. Based on the feedback, we have updated the draft highlighting our changes with blue text. Driven by your feedback we have conducted additional experiments with results in Figure 3 (per-token detection performance), Figure 4 (open-source generation detection),  Figure 9 (detecting different prompting strategies), and Figure 17 (ablation). Below is our response with a point-by-point note for the questions/weaknesses listed.
>
> __1. Zero-shot Detection Claim.__
> - We would like to clarify that we do not optimize the threshold for the target task (in all cases/experiments), but use a global threshold obtained from Binoculars. As mentioned in section 4.3: since our global threshold is tuned on a ChatGPT benchmark dataset (News, Creative Writing, Essay) by [1] (Ghostbusters method), we make an exception "to be sure that we meet the OOD criteria, we do not include the ChatGPT datasets in the threshold determination, and only use samples from CC News, CNN, and PubMed (generated via LLaMA and Falcon) to determine threshold to do predictions/evaluation on ChatGPT benchmark dataset (Figure 1, 2, & 3). For open-source datasets (CC News, CNN, and PubMed), we report AUC plots for all methods that are threshold-invariant for all methods (Figure 4). With this OOD threshold, we demonstrate that reliably detects texts generated by different LMs and from different domains.
>
>
> __2. Roberta, LR GLTR, Stylistic, Nela not explained.__
> - Thank you for pointing out, that we have added a note in the M4 Datasets note in section 5.1
>
> __3. Choice of baselines and across experiments__
> - In this work, our emphasis is directed toward baselines that function within post hoc, out-of-domain (zero-order), and black-box detection scenarios. We use state-of-the-art Ghostbusters [1], GPTZero, and DetectGPT to compare detection performance over various datasets in section 4.
> - In section 5, we evaluate the reliability of Binoculars in various settings that constitute edge cases and
> interesting behaviors of our detector (multi-domain, languages, non-native English text, etc.)
> - We updated the draft to mention our choice of baselines and rationale towards the end of Section 2.
>
> __4.  Methodology Results of M4 Datasets [2]__
> - From Wang et al., we use ChatGPT vs Human performance table. To aid OOD comparison between Binoculars and other baselines, we use the mean of OOD-reported performances by Wang et al. For each baseline, for the "arXiv" domain for example, we use the mean of performance when the respective baseline is trained on {Wikipedia, WikiHow, Reddit, ELI5, PeerRead}.
> - We update the caption Figure 7 to clarify this.
>
> __5. Absence baseline comparison on M4 dataset__
> - Rationale: As mentioned above, in Section 4 we use baseline methods to compare our performance with established zero-shot, OOD domain detectors on various known benchmark datasets. Our intent for M4 datasets is to study our detector's performance in multiple domains, languages, and text generators and report non-relative claims.
> -  Constraint: Unlike other feedback, we could not run other methods on M4 datasets which consist of 39 datasets with 3k samples each due to sizable constraints: a) Ghostbusters require approx $1150 OpenAI credits to get log-probs for 46 million tokens (512 per sample), b) computational constraint for DetectGPT which requires computing multiple perturbations.
> - Thus, we move M4 datasets experiments to Section 5 (reliability).
>
> __6. Baselines in Orca Dataset__
> - We run other methods to append baselines in Figure 9 to show our method is more robust than other methods.
>
> __7. Inconsistency/typos in formulation/notation.__
> - We thank you for the detailed feedback and have updated to fix all pointers.
>
> __8. Intuition on x-PPL, its interpretation, and its effect on the detection method__
> - __Lower/Upper Bound:__ Binoculars score are ratios of two cross-entropy measures (both being positive) and is a positive number
> - __Impact of using identical models for scoring__: We perform this experiment and report the findings in Figure 17 in Appendix A5. We observe using Falcon-7B as $M_1$ and $M_2$ comes close to Binocular's performance.
>
>
>
> [1] Vivek Verma, Eve Fleisig, Nicholas Tomlin, and Dan Klein. Ghostbuster: Detecting Text Ghost-
> written by Large Language Models. arxiv:2305.15047[cs], May 2023. doi: 10.48550/arXiv.2305.
> 15047. URL http://arxiv.org/abs/2305.15047.
> [2] Yuxia Wang, Jonibek Mansurov, Petar Ivanov, Jinyan Su, Artem Shelmanov, Akim Tsvigun,
> Chenxi Whitehouse, Osama Mohammed Afzal, Tarek Mahmoud, Alham Fikri Aji, and Preslav
> Nakov. M4: Multi-generator, Multi-domain, and Multi-lingual Black-Box Machine-Generated
> Text Detection. arxiv:2305.14902[cs], May 2023. doi: 10.48550/arXiv.2305.14902. URL
> http://arxiv.org/abs/2305.14902

---

> ### Author Response · Authors · 2023-11-20
> **Response to Reviewer FmjG (Part 2)**
>
> __9. Use of scoring models trained on different domains and other aspects of Binoculars.__
> - Please note, by our x-PPL definition, we have an implicit constraint of having to use two models to share a tokenizer, and most available open-source models have a high degree of training set overlap (eg. Falcon, Llama family).
> - The cross-perplexity measures the degree to which two models' next-token distributions overlap. We suspect cross-ppl from two models trained on dissimilar datasets would provide more noise than the signal we want to detect (see updated text section 3.3 for motivation of x-PPL). The similar behavior of the transformers model is what we depend on for cross-ppl and Binoculars score definition. Thank you for mentioning this important point. We expanded the definition section to mention the impact of different training sets and will be happy to provide any more information required.
> - In the ablation experiment (in Figure 17), we see how x-PPL and PPl alone aren't enough for reliable detection performance.
> - We are happy to answer any further or follow-up questions on this.
>
> __10. When showing FPR/TPR plots, how are these plots generated? by changing the detection threshold? I do not understand why you say 0.01% FPR threshold is "The smallest threshold we can comprehensively evaluate with our limited compute resources."__
> - Yes, for each method we move the threshold to achieve 0.01% FPR and report the TPR figures to compare performance in low false positive regime.
> - The FPR rate we can achieve on a log scale is a function of the number of samples. For all our experiments, we use a balanced dataset (50-50 human and LM samples). At 0.01% FPR, we would need 10k samples per class, and at 0.001% we would need 100k samples per class.
>
> To summarize, we present a novel detector that is able to maintain high efficacy in a false-positive regime under various domains/datasets/baselines, using only open-source components to detect in zero-shot settings. Broadly, we highlight the need to evaluate detectors in terms of low-false positive metrics and showcase how our method is performative in this regime (note: in principle, by lowering the threshold even further we can virtually eliminate any false positives). We further study our detector’s reliability in settings that constitute edge cases to understand interesting behaviors, abilities, and limitations of our detector.
>
> We are happy to take on any follow-up questions or other feedback.
>
> Again, we thank our reviewers for their time and contribution to this key area of secure and safe machine learning in the generative AI era.

---

> > ### Author Response · Authors · 2023-11-21
> > **A Gentle Reminder**
> >
> > Dear Reviewer FmjG,
> >
> > As the discussion period ends, we would like to kindly ask for your reply to our remarks. We feel we have addressed your concerns and we look forward to further discussions and guidance. Please let us know if there is anything else we can do or other questions we can answer.

---

### Meta-Review · Area_Chair_z3zk · 2023-12-05

**Metareview:**

This paper describes an approach to LLM-generated text detection. The approach is called *Binoculars* due to its use of two models and contrasting their perplexity and thus leveraging the metaphor of examining text through *two lenses*. The authors define a *cross-perplexity* measure that captures the perplexity of an LLM when examining output of another. This is used to normalize the perplexity measure of one of the LLMs to derive the *Binocular Score* in Eq. (4).

This paper has two main strengths:
+ The simplicity of the proposed approach. The entire approach is effectively captured in Eq. (4) of the main paper and is entirely based on perplexity measures. The reliance on pretrained LLMs without the need for finetuning is a plus.
+ Good performance on a broad variety of benchmarks and benchmark scenarios.

The main negative points, however, outweigh these positives:
+ Problems with typos and general presentation issues. Reviewers pointed to problems with organization, notation, missing definitions, and a variety of other significant issues with the submitted manuscript.
+ Overall lack of motivation and analysis for *why* the Binoculars model works. The submitted manuscript provides no intuitive or theoretical motivations for why the Binoculars Score should be a good metric for LLM-generated text. No qualitative experiments were performed to probe deeper aspects of the approach. It seems that the paper relies almost entirely on the good empirical results reported, which -- especially for such a simple approach -- is not enough to meet the bar for acceptance at ICLR.
+ Missing descriptions of baselines and inconsistencies between baselines considered in different plots.

Many of the issues raised by reviewers were partially addressed by the authors in the *revised* version of the manuscript, however this practically constitutes a *Major Revision* of the work performed in rebuttal. In fact, the AC notes that the revised manuscript is **now ten pages long**, a full page over the nine-page submission limit. This confirms that the submitted manuscript was not mature enough to merit publications and that the manuscript is in need of more rounds of revision.

**Justification For Why Not Higher Score:**

The consensus of the majority of reviewers was clear from the initial reviews. All four reviewers pointed to problems with organization, notation, missing definitions, confusing use of baselines, and a variety of other significant issues with the submitted manuscript. The positive opinion of one out of four reviewers is not enough to outweigh the specific points made be the other three.

**Justification For Why Not Lower Score:**

N/A

---

### Decision · Program_Chairs · 2024-01-16

Reject